# Evaluation of Soil–Structure Interaction in Structure Models via Shaking Table Test

Seongnoh Ahn, Gun Park, Hyungchul Yoon, Jae-Hyeok Han and Jongwon Jung *

School of Civil Engineering, Chungbuk National University, Cheongju 28644, Chungbuk, Korea; copoi2212@chungbuk.ac.kr (S.A.); silvist@g.cbnu.ac.kr (G.P.); hyoon@cbnu.ac.kr (H.Y.); aksgdlsksp@naver.com (J.-H.H.)
* Correspondence: jjung@chungbuk.ac.kr

**Abstract:** Modeling the soil–structure interaction (SSI) in seismic design involves the use of soil response curves for single-degree-of-freedom (SDOF) structures; however, real structures have multiple degrees of freedom (MDOF). In this study, shaking-table-derived p-y curves for SDOF and MDOF superstructures were compared using numerical analysis. It was found that an MDOF structure experienced less displacement than an SDOF structure of the same weight, but the effect of increasing the DOF decreased at greater pile depths. Numerical analysis results estimated using the natural periods and mass participation rates of the structures were similar to those of shaking table tests. Abbreviations: finite element: FE; frequency response function: FRF; multiple degrees of freedom: MDOF; single degree of freedom: SDOF; soil–structure interaction: SSI.

**Keywords:** shaking table test; soil–structure interaction; p-y curve; pile foundation; seismic design

## 1. Introduction

In the past 10 years, approximately 1500 earthquakes of magnitude 5 or greater and approximately 10 earthquakes of magnitude 7 or greater have occurred worldwide each year [1]. In addition to structural deformation and destruction, earthquakes cause damage to humans and nonstructural property. To prepare for these events, seismic design is important for the construction of structures that can withstand earthquake loading as defined under certain regulations [2]. In the seismic design of a structure, the effect of the superstructure, foundation, and site under the foundation should be considered together. In pile-foundation seismic design, equivalent static analysis is generally used to determine the lateral load (seismic load) on the pile by constructing a p-y curve that includes the nonlinear behavior of the ground [3]. A p-y curve shows the nonlinear relationship between the displacement of a pile (y) caused by the lateral load and the reaction force (p) of the ground, which is modeled by describing the ground as a set of springs for which the change in the spring coefficient is dependent on the depth and compaction condition of the soil [4]. As the p-y curve changes based on the condition of the ground, it is difficult in practice to calculate it for all conditions; therefore, many researchers have proposed the use of p-y curves obtained under various ground and load conditions to produce a representative curve [5–8]. Because most approaches assume a condition in which a cyclic or static load to is applied to the pile head, they are able to explain the nonlinear relationship between the pile displacement and the ground. However, because the stiffness of soil decreases as the amplitude of the load increases, and because there are cases in which the inertia or attenuation of the ground that occurs under dynamic loading cannot be evaluated [9], dynamic loads such as earthquake loads cannot be reasonably considered in seismic design [10–12]. To address this issue, studies attempting to develop a dynamic p-y curve for piles are currently underway. Shaking table tests used to reproduce actual earthquake effects use models that have similitude to real field structures [13,14]. Kim et al. (2018) demonstrated that the dynamic p-y curve changes with the position of the pile cap

after it has been combined with a single-degree-of-freedom (SDOF) weight [4]. Lim and Jeong (2017) proposed a p-y curve for sandy soil based on a division of the combining condition of the single pile and weight into two conditions: a hinge condition and a fixed condition [10]. Because the stress-deformation behavior in terms of, for example, confining pressure depends on the indoor experimental condition, the stress-deformation of a real construction site cannot be completely described through model similitude alone; to address this, studies in which current approaches are supplemented by centrifugal model experiments are being conducted [15–17]. In addition to experimental derivation, p-y curves based on numerical analysis have also been proposed. Choi and Ahn (2020) numerically derived the interference effect and change occurring in the p-y curve of a group pile under seismic loading [18]. Hokmadadi et al. (2014) assessed the validity of the numerical analysis approach by comparing the results obtained from model shaking table testing and numerical analysis [19]. Ihsan and Tahsin (2019) compared the results of experimental and numerical analysis of piles subjected to passive loading and revealed that the numerical results overestimated deflection but produced reasonably consistent bending moments [20]. Although more accurate p-y curves based on studies of soil–structure interaction (SSI) under dynamic load conditions have been proposed, such studies have assumed that the superstructure is an SDOF structure [4,10,21]. In practice, however, a field structure will be a multiple-degrees-of-freedom (MDOF) structure that will likely produce a p-y curve that differs from that of an SDOF structure. Therefore, in this study we conducted shaking table model tests in which the structure was modeled first with no superstructure, and then with SDOF and MDOF structures with the same weights to obtain the dynamic behavior of the pile in the presence or absence of a superstructure and, in the process, compare the SDOF and MDOF structure cases. In addition, numerical analysis of the structure was performed. The test results and analysis results were compared, and the cause of the difference according to the model of the superstructure was analyzed. The results of the shaking table test will improve our knowledge of the dynamic behavior of the real structures.

## 2. Model Test

### 2.1. Shaking Table Test

A pile foundation is affected not only by vertical loads from the superstructure but also by lateral dynamic loads produced by the interaction of the superstructure with seismic and wind loads [22,23]. By applying a hydraulic pump to a shaking table test device to which a structure or soil–structure model is attached, the behavior of the structure under seismic loading can be assessed. In this study, a model shaking table experiment involving SSI was conducted. An STC-V101 shaking table device with a 1000 mm wide × 1000 mm long plate that could control a maximum acceleration and displacement of 1 g and ± 100 mm, respectively, was used to carry out the experiment. An acrylic plate sandbox comprising nine layers, each with the dimensions (h × w × l) 55 × mm × 600 mm × 600 mm, was attached to the shaking table. Springs were attached to each layer of the acrylic plate to reduce the amplitudes of the reflected waves generated by the seismic loads impacting the sandbox wall (Figure 1).

### 2.2. Test Model Similitude

Testing was conducted by reducing the ground and structures of the model. Because a 1 g shaking table test device cannot accurately recreate the confining pressure of the ground occurring at a real site [10], the model structures were designed to ensure a degree of similitude with actual structures, with specific rules applied to improve the applicability of the model's results to the field. A special case in which $\lambda_p = 1$ and $\lambda_\varepsilon = \lambda^{0.5}$ was applied under the assumption that the shear wave velocity was an unknown similitude condition for the 1 g shaking table test proposed by Iai in 1989 [13]. Despite this attempt to ensure similitude, however, the degree to which the scaling factors for the stiffness and thickness of the model pile could be satisfied was limited. In pile foundations affected by lateral load,

the bending stiffness significantly influences the pile behavior; thus, the model structure was manufactured by satisfying the scaling factor for bending stiffness even though the scaling factor for thickness itself could not be satisfied. For the shaking table test, a scaling factor of 26.5 was used; the conditions of similitude for the model are summarized in Table 1. The prototype structure pile structure was a concrete pile with a diameter of 500 mm, a thickness of 100 mm, and an elasticity modulus of 14 GPa. The single pile used in the model structure was fabricated from aluminum and had a diameter of 12 mm, a thickness of 2 mm, and a length of 490 mm, with the dimensions all suitable to the height of the sandbox. Based on a previous study on the uniaxial strain value [4], the modulus of elasticity was set at 72.5 GPa.

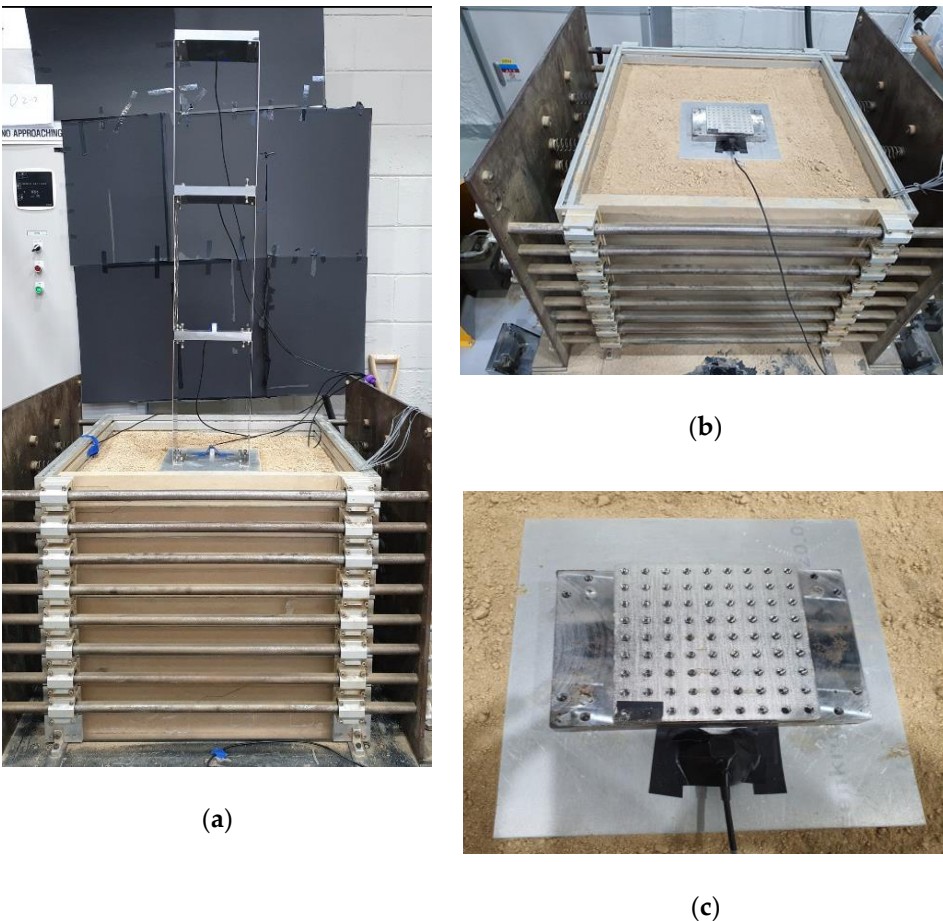

**Figure 1.** 1 g shaking table test device setup. (**a**) MDOF structure model, (**b**) SDOF structure model, (**c**) Zoomed SDOF superstructure.

**Table 1.** Scaling factor and pile properties.

| Classification | Scaling Factor [13] | Model Pile | Prototype Pile |
|---|---|---|---|
| Material | n/a | Aluminum 6061 | Concrete |
| Pile length (cm) | λ | 49 | 1298.5 |
| Diameter (cm) | λ | 1.2 | 50 |
| Thickness (cm) | λ | 0.2 | 10 |
| Elastic modulus (GPa) | λ | 72.5 | 27 |

### 2.3. Pile and Structure Properties

The behavior of a pile foundation depends on the properties of the long, middle, and short piles [10,24–26]. In this study, the pile analysis conditions used by Broms in 1964 were referred to (Table 2) [27]. To satisfy the long pile condition, the coefficient of horizontal

subgrade reaction in sandy soil was assumed to be 20,000 kN/m$^3$, based on the value proposed by Davisson in 1970 [28]. The file foundation comprised a single pile, which, to simulate a rock-socketed pile, was fixed to an aluminum plate at the bottom of the sandbox. In consideration of the loading of the model structure, the pile cap was an aluminum plate with a width of 245 mm, a length of 245 mm, and a thickness of 20 mm. The testing was performed with the pile head fixed.

**Table 2.** Pile analysis conditions [27].

| Classification | Sand | Clay |
|:---:|:---:|:---:|
| Short pile | $\eta L < 2.0$ | $\beta L < 2.25$ |
| Immediate pile | $2.0 \leq \eta L \leq 4.0$ | - |
| Long pile | $\eta L > 4.0$ | $\beta L > 2.25$ |
| $\beta = \sqrt[4]{\frac{K_h D}{4EI}}, \eta = \sqrt[5]{\frac{\eta_h}{EI}}$ | | |

The structure above the pile cap produces a mass-dependent inertial force when it is affected by a dynamic lateral load. This inertial force generates kinetic energy that is propagated as a wave back to the ground, making it necessary to consider the inertial force when performing SSI structure analysis. Previous studies have proposed that the p-y curve can be generated by modeling the pile cap and superstructure as an SDOF structure to which a dynamic load was applied. However, real structures are not SDOF structures but rather MDOF structures with specific shapes, and are therefore difficult to accurately assess based on tests conducted with SDOF structure models. In this study, models without a structure (W/O structure) and with three-story SDOF and MDOF structures were tested. The SDOF and MDOF structures had the same weight and were used to compare the differences in structure mode (Figure 2).

### 2.4. Types and Locations of Sensors

The behavior of the structures during the shaking table tests was measured using eight strain gauges attached to the pile and one linear variable differential transformer attached to the shaking table plate. The bending moment that occurs at each depth is dependent on the pile condition [29]. In the case of a fixed long pile head, the bending moment converges to zero at the tip of the pile and changes in direction at the top and bottom of the pile [20]. Therefore, in our tests, the strain gauges of the pile were attached as shown in Figure 3 to determine the magnitudes of the bending moment and the location of bending moment at which the direction of the moment changed.

### 2.5. Test Soil

The soil used in this study was weathered residual soil collected from the field. The properties of the soil are listed in Table 3. Using the unified classification method, the soil was classified as sand with poor particle size (SP). The 0.18 m$^3$ sandbox was filled with soil to a density of 1.67 t/m$^3$ using impact compaction. As the strength of the clay component in the soil increased over time, the test was conducted immediately after the soil was compacted.

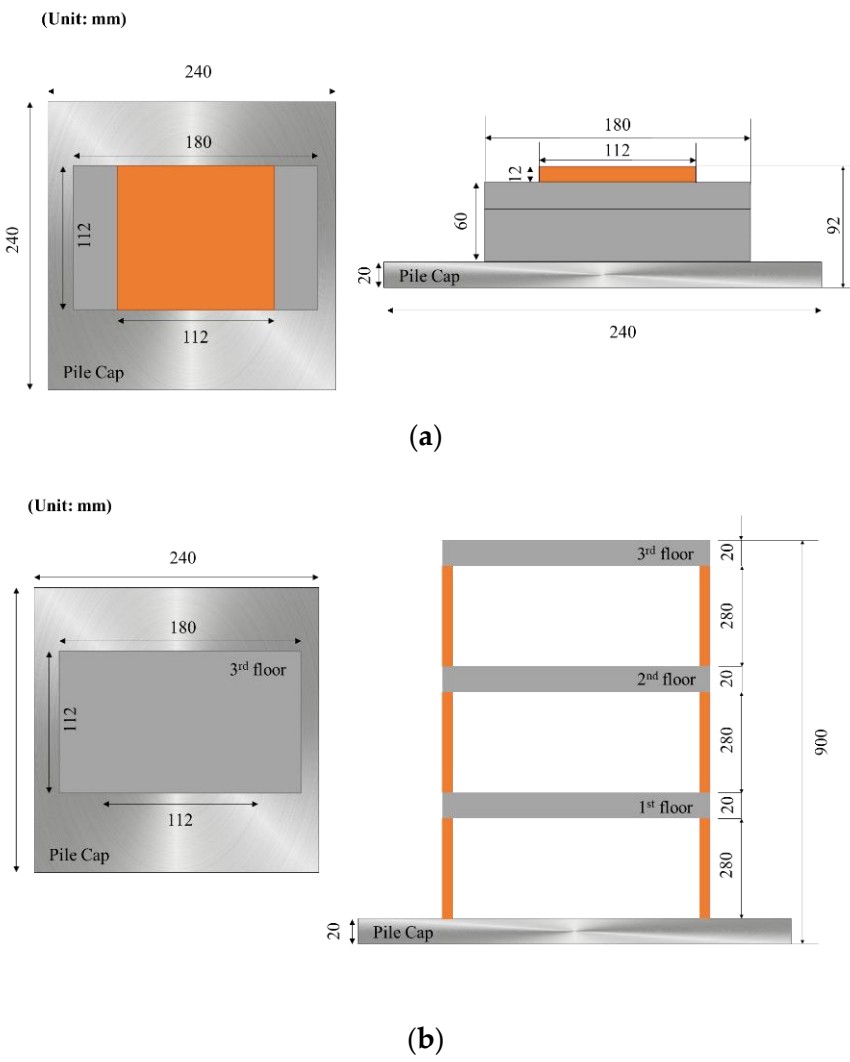

**Figure 2.** Model structure shapes. (**a**) SDOF structure model, (**b**) MDOF structure model.

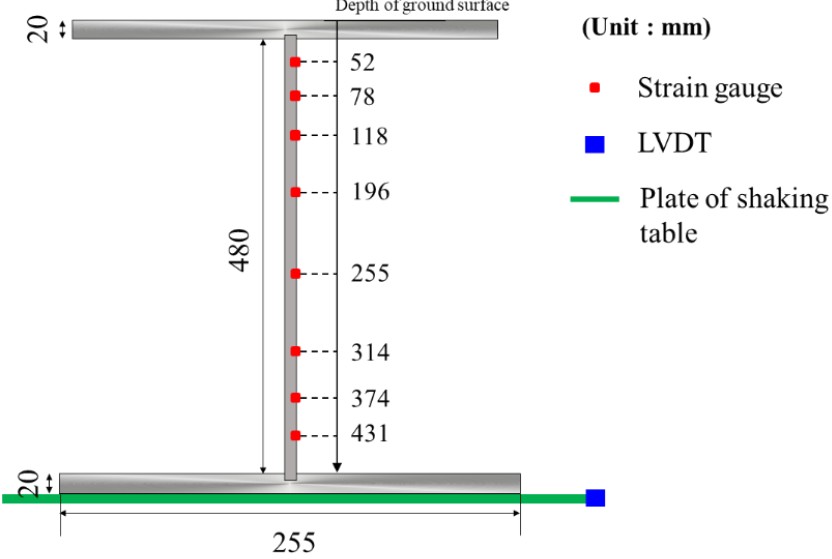

**Figure 3.** Locations of sensors used for shaking table test.

**Table 3.** Weathered residual soil properties.

| Index | | Sample |
|---|---|---|
| | $D_{10}$ | 0.17 |
| | $D_{30}$ | 0.47 |
| | $D_{60}$ | 1.1 |
| Sieve analysis | $C_u$ | 6.47 |
| | $C_C$ | 1.18 |
| | Passing No. 4 sieve (%) | 100 |
| | Passing No. 200 sieve (%) | 2.42 |
| USCS | | SP |
| Specific gravity | | 2.59 |
| Unit weight (g/cm$^3$) | | 1.67 |

### 2.6. Test Program Settings

The shaking table test apparatus excites the shaking table plate by applying a vibrational load through a hydraulic pump; this loading is read as displacement data; however, to enable dynamic analysis, the displacement data were differentiated twice and expressed as acceleration. The test loads were applied in sine wave form to enable easier determination of the desired size, frequency, and magnitude. The frequency and acceleration were varied for each case (Table 4) over an acceleration magnitude range of 0.3–0.5 g; for the SDOF superstructure case, testing was only performed up to 0.4 g to avoid exceeding the bending stress of the aluminum pile. To assess the effect of input frequency on the SSI, the frequency was varied from 3 to 10 Hz at each acceleration level. The 2–10 Hz section is an important factor in indicating the effective ground acceleration [30]. Due to the displacement limitations of the shaking table test, testing was carried out from 3 Hz.

**Table 4.** Shaking table test program settings.

| Case | Input Acceleration (g) | Input Frequency (Hz) |
|---|---|---|
| W/O structure | 0.3, 0.4, 0.5 | 3–10 |
| SDOF structure | 0.3, 0.4 | 3–10 |
| MDOF structure | 0.3, 0.4, 0.5 | 3–10 |

### 2.7. Dynamic p-y Curves

Because piles have less resistance to lateral than to vertical loading, lateral displacement occurs even under relatively small lateral loading. Therefore, the allowable lateral displacement is usually dominant in designing a pile. The soil subgrade reaction method is used to calculate the displacement and moment generated in a pile by modeling it as a beam under the Winkler foundation model, which assumes that the ground is supported by multiple springs and that the piles are beams. The Winkler equation is as follows:

$$E_p I_p \frac{d^4 y}{dz^4} - p \frac{d^2 y}{dz^2} - p = 0. \tag{1}$$

To construct a dynamic p-y curve, it is necessary to obtain a bending moment shape for each pile depth. Using data obtained from a strain gauge, the bending moment can be calculated as follows:

$$M = \frac{EI \cdot \epsilon}{y}, \tag{2}$$

where $EI$ is the flexural rigidity of the pile, $\epsilon$ is the strain of the pile, and $y$ is the distance to the center of the pile. By taking the second derivative of the bending moment, the ground

reaction force (*p*) on the pile can be obtained; similarly, the displacement (*y*) of the pile can be expressed by double integration of the bending moment:

$$p = \frac{d^2}{dz^2} M(z), \tag{3}$$

$$y = \iint \frac{M(z)}{EI} dz. \tag{4}$$

## 3. Numerical Analysis

*Finite Element Models*

Using the 1 g shaking table apparatus, p-y curves were obtained for each of the three superstructure modes in sandy soil. Using the results of the shaking table tests described in Section 2.6, LS-DYNA, a general-purpose finite element (FE) program, was used to perform numerical analysis to obtain the p-y curves in terms of the characteristics of the ground and the mode of each superstructure, the natural period of each structure, and the SSI via response analysis. The model types used for the respective LS-DYNA renderings are shown in Figure 4.

The FE models applied to the analysis comprised the ground, pile and a mass-type, two-story, or three-story structure. The ground, structure foundation, and mass structure were generated using eight-node solid elements, whereas the pile and two- and three-story structures were generated using four-node shell elements. Seismic load was applied to the bottom of soil which was assumed to roller (the degree of freedom in the vertical were constrained, and other degree of freedom were free), and time-history analysis was conducted. A spring/damper was applied to the surface of the pile and to the foundation of the structure in contact with the ground to model the effects of the SSI. Both the structure and the ground were assumed to be elastic; the material properties, element type, and number of elements applied in the analysis are listed in Tables 5 and 6.

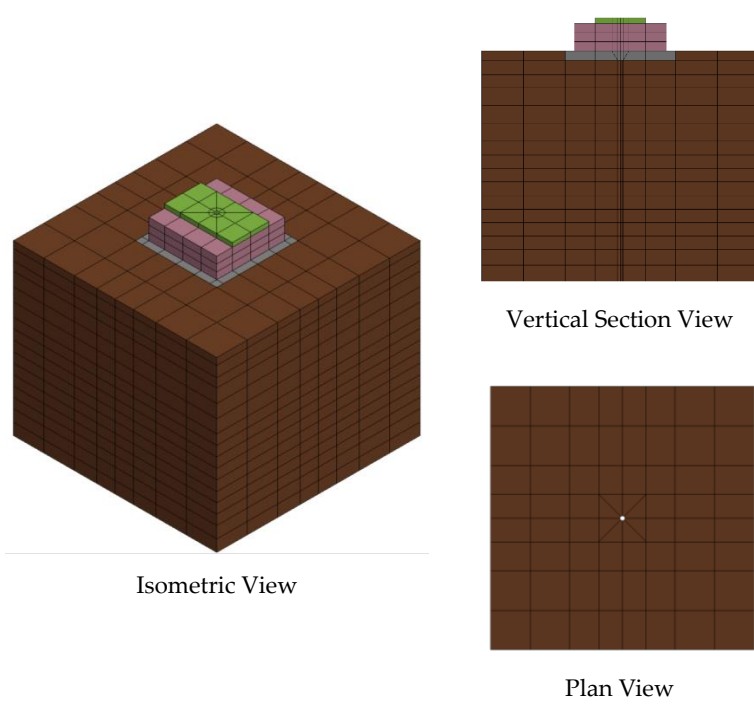

Isometric View

Vertical Section View

Plan View

(**a**)

**Figure 4.** *Cont.*

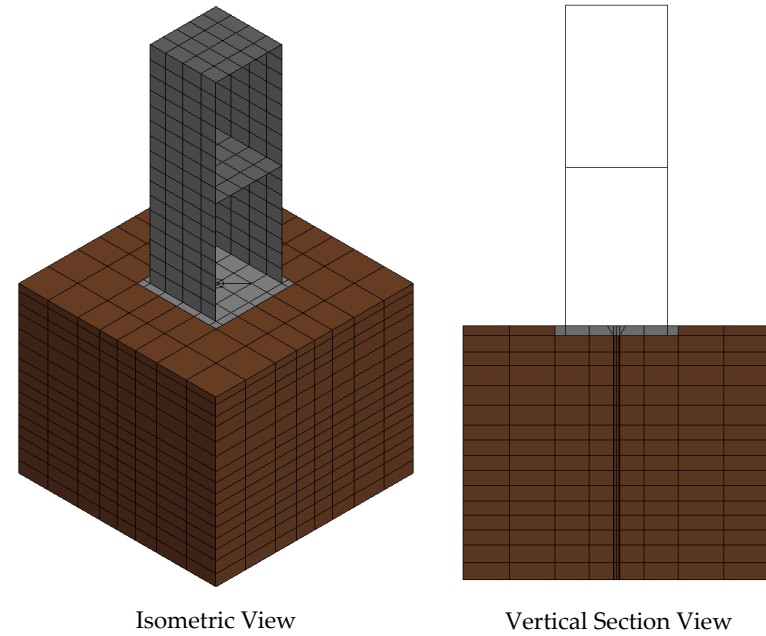

Isometric View          Vertical Section View

(**b**)

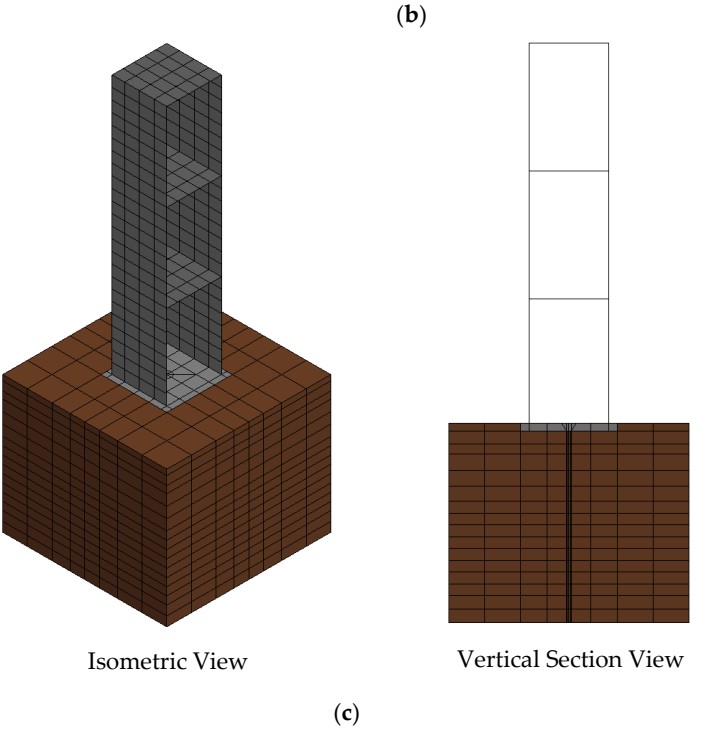

Isometric View          Vertical Section View

(**c**)

**Figure 4.** Models for different structure types. (**a**) Mass-type, (**b**) two-story building type, (**c**) three-story building type.

**Table 5.** Material properties used in FE models.

| Material | Modulus of Elasticity (N/mm$^2$) | Poisson's Ratio | Unit of Weight (ton/m$^3$) |
|---|---|---|---|
| Soil | 106.40 | 0.33 | 1.600 |
| Structure/Mass structure | 210,000.00 | 0.30 | 7.850 |
| Pile | 68,900.00 | 0.33 | 2.700 |

**Table 6.** Element type and number of elements applied in the analysis.

| | | Element Type | Number of Element |
|---|---|---|---|
| Mass type | Mass | Solid (8 node) | 120 |
| | Basement | Solid (8 node) | 104 |
| | Pile | Shell (4 node) | 120 |
| | Soil | Solid (8 node) | 1068 |
| Two-story building type | Structure | Shell (4 node) | 168 |
| | Basement | Solid (8 node) | 104 |
| | Pile | Shell (4 node) | 120 |
| | Soil | Solid (8 node) | 1068 |
| Three-story building type | Structure | Shell (4 node) | 252 |
| | Basement | Solid (8 node) | 104 |
| | Pile | Shell (4 node) | 120 |
| | Soil | Solid (8 node) | 1068 |

The elastic modulus of the soil used in the analysis was calculated based on a resonant column test; the material properties applied to the structures and piles reflected the material properties of the steel and aluminum, respectively, used in the experimental assessment.

## 4. Results and Discussion

### 4.1. Bending Moment and Pile Displacement Distribution According to Depth

Figure 5a shows the distribution of the bending moment by depth at the point at which the maximum bending moment appeared in the pile, as determined by the shape of the superstructure. Relative to the W/O structure case, the SDOF and MDOF structures increased the maximum bending moment generated in the pile by factors of approximately 1.85 and 1.16, respectively. We attribute this disparity in the amount of increase to the fact that the amplification induced by the inertial force and natural period varied depending on the mode of the structure independently of the weight of the structure. The inflection point of the bending moment appeared within the lower 400 mm of the pile, with the bending moment occurring in the positive (+) direction at approximately 250 mm (where 0 mm indicates the pile top), which was the central region of the pile. The maximum bending moment occurred at the ground surface; this was also true for the MDOF structure, whereas in the SDOF structure the maximum occurred at a depth of approximately 120 mm. Previous research results have shown that the change in moment decreases at the ground surface, likely as a result of the change in the confining pressure of the upper ground depending on whether or not a pile cap is present [3,11,31]. Figure 5b shows the displacement of the pile by depth at different input accelerations. As the acceleration increased, the degree of displacement by depth increased. In each case, the maximum displacement occurred at the ground surface (0 cm), with magnitudes of 4, 5.5, and approximately 6 mm at 0.3, 0.4, and 0.5 g, respectively.

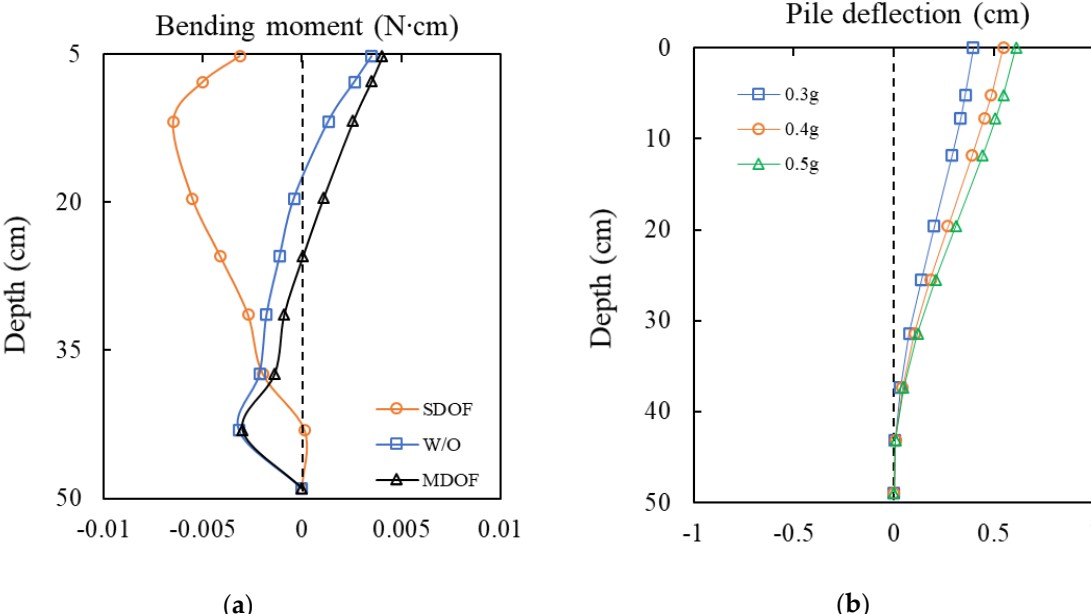

**Figure 5.** Bending moment and pile deflection according to depth. (**a**)Bending moment by the superstructure: input frequency = 7 Hz; Acc = 0.4 g, (**b**) Pile deflection by the input acceleration: input frequency = 7 Hz; superstructure = W/O structure.

### 4.2. Natural Period According to Model Soil and Structure Mode

Figure 6a shows p-y curves produced by the W/O structure at a depth of 11.8 cm from the ground surface at different input frequencies with an input acceleration of 0.4 g. The largest displacement and soil resistance values occurred at 7 Hz, likely because the input and natural frequencies of the soil–pile foundation were in resonance. The soil resistance increased more when the natural frequency was larger than it did when the natural frequency was small; however, the displacement decreased and the p-y curve slope increased. Figure 6b shows the soil resistance at the ground surface (0 cm) at different input frequencies for the SDOF and W/O structures. For the SDOF structure, the soil resistance was higher at all frequencies than it was for the W/O structure. The maximum ground reaction force to the SDOF structure occurred at 8 Hz, at which point it was approximately five times that on the W/O structure. In the low-frequency range (3–6 Hz), the ratio of magnitudes was constant at approximately 3.4. We attribute this to the increase in the inertia force on the SDOF structure through the application of additional loading. By contrast, the MDOF structure experienced no consistent increase in soil resistance relative to the W/O structure over the entire 3–10 Hz range (Figure 6c). We attribute this to the canceling of different inertial forces as the number of structural modes increased to three. Nevertheless, the soil resistance to the MDOF structure rose sharply (by a factor of approximately 1.4) relative to that of the W/O structure in the region of 5 Hz. However, at 7 Hz—the measured natural period of the ground—it declined to approximately 0.9 that of the W/O structure. We attribute this to an effect of the structural period and discuss it in more detail in Section 4.5.

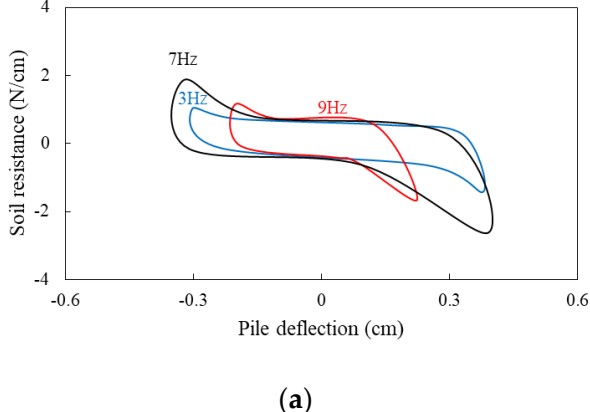

(**a**)

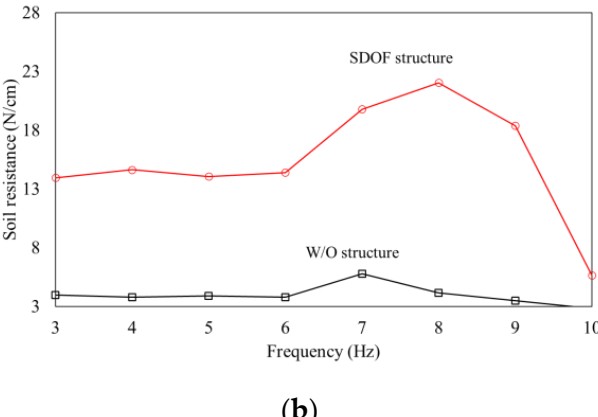

(**b**)

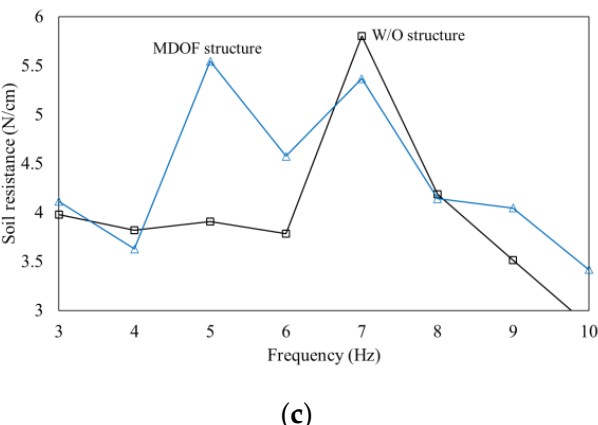

(**c**)

**Figure 6.** Soil resistance to different structures at different input frequencies. (**a**) p-y curves for W/O structure at depth = 11.8 cm, acc. = 0.4 g. (**b**) Comparison of soil resistances by frequency for SDOF and W/O structures. (**c**) Comparison of soil resistances by frequency for MDOF and W/O structures.

### 4.3. Soil–Structure Interaction Dynamic p-y Curve

The measured maximum values of soil resistance (p) and pile displacement (y) at different depths measured by the strain gauge attached to the pile are shown in Figure 7. For the W/O structure model (Figure 7a), the slope of the curve increased as the depth increased to 7.8, 11.8, 37.4, and finally 43.1 cm, although a slope similar to that at 37.4 cm occurred at a depth of 0 cm. For the SDOF structure model (Figure 7b), the largest slope occurred at a depth of 43.1 cm; in addition, the slope was larger at 7.8 cm than it was at

11.8 cm. For the MDOF structure model (Figure 7c), the slope was larger at a depth of 7.8 cm than it was at 11.8 cm, and the slopes at 0 and 37.4 cm were similar.

(**a**)
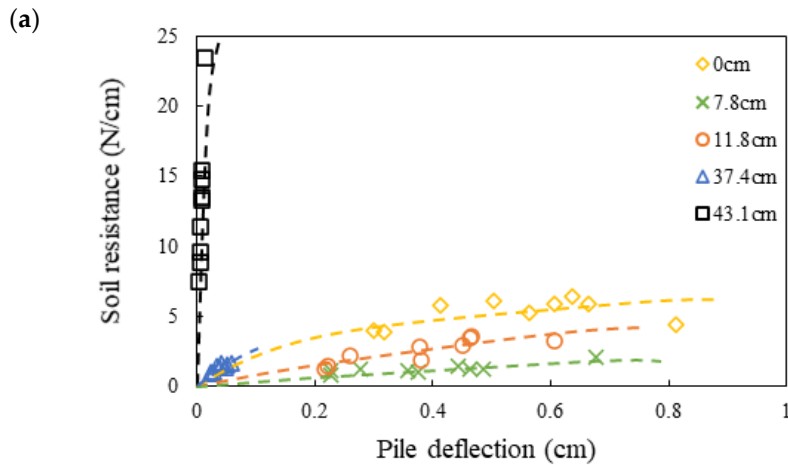

(**b**)
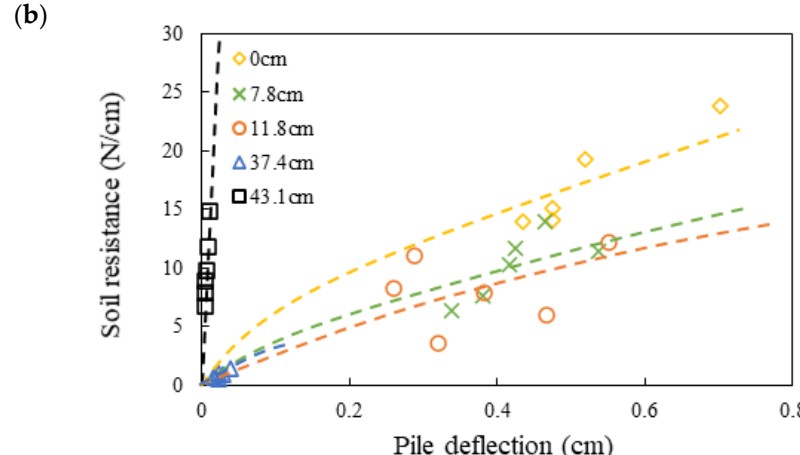

(**c**)
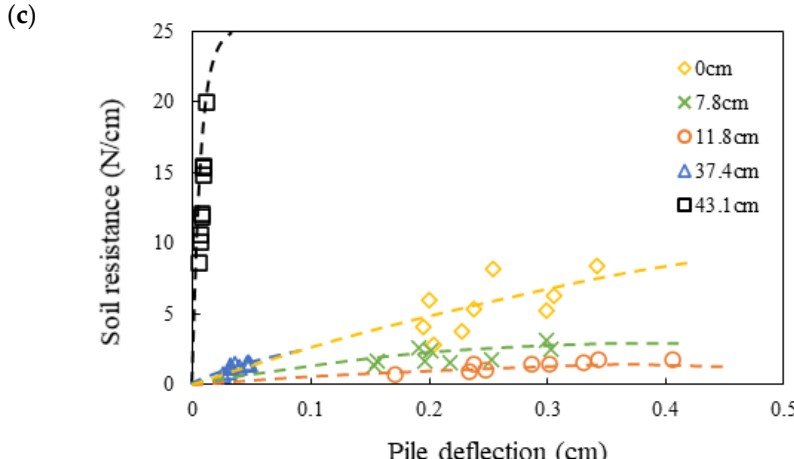

**Figure 7.** Dynamic p-y backbone curves for (**a**) W/O, (**b**) SDOF, and (**c**) MDOF structure models.

The p-y curves for each superstructure case at four depths are compared in Figure 8. At the surface (0 cm; Figure 8a), the slopes of the curves increased in order from the W/O to the MDOF to the SDOF cases. At 7.8 cm (Figure 8b), the ordering by slope was the same

but, in each case, the absolute degree of rise was reduced relative to the 0 cm case. As the depth increased beyond a certain level (Figure 8c), the differences between the slopes of the p-y curves for the respective superstructures gradually decreased until the curves nearly converged near the tip of the pile (Figure 8d).

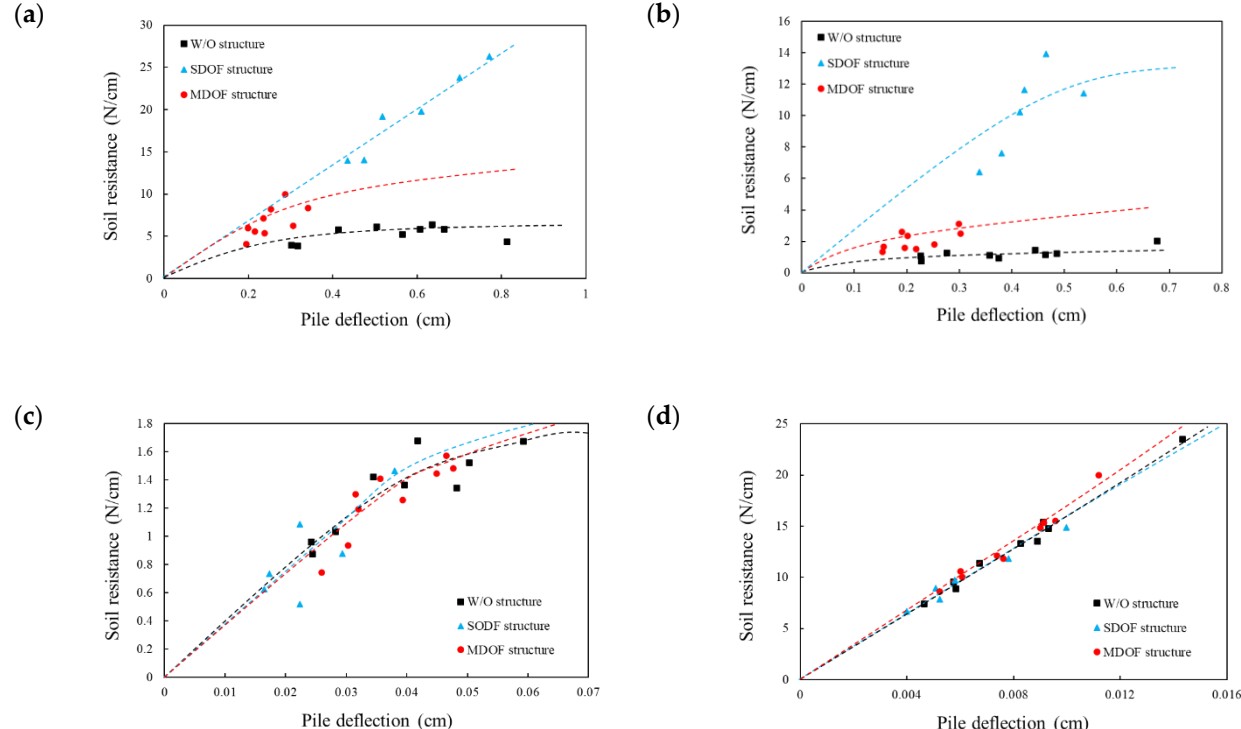

**Figure 8.** Dynamic p-y backbone curves for different superstructures at depths of (**a**) 0, (**b**) 7.8, (**c**) 37.4, and (**d**) 43.1 cm.

### 4.4. Validation Analysis

To confirm the accuracy of the numerical analysis results, the time-varying load shown in Figure 9 was applied to the models described in Section Finite Element Models to obtain experimental results for pile displacement by height (Figure 10).

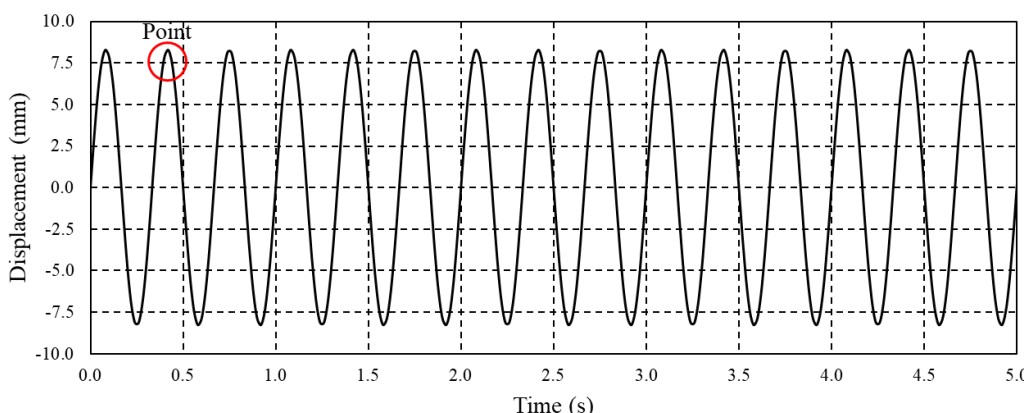

**Figure 9.** Time-varying load applied in numerical analysis.

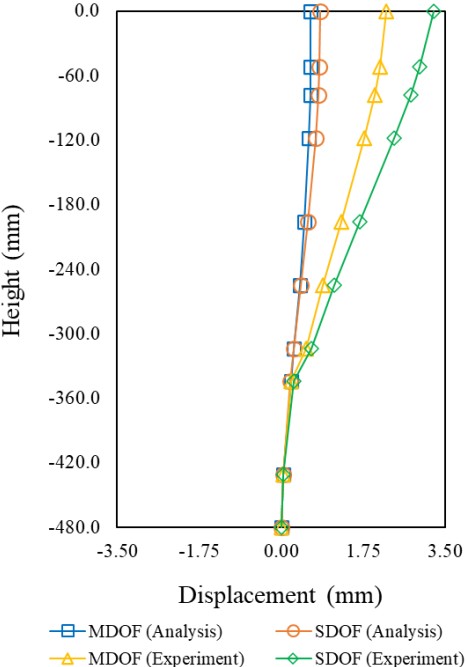

**Figure 10.** Comparison of experimental and analysis results for SDOF and MDOF structures: displacement of pile by height.

It can be observed that, in both the test and analysis results, the SDOF structure underwent a larger displacement than the MDOF structure. However, both structures showed maximum displacements smaller than those determined from the experimental results. Table 7 shows the displacement at each layers of the piles estimated by the analysis and experimental results.

**Table 7.** Displacement at each layers of the piles.

| Height (mm) | SDOF | | | MDOF | | |
|---|---|---|---|---|---|---|
| | Experiment (mm) | Analysis (mm) | Difference (%) | Experiment (mm) | Analysis (mm) | Difference (%) |
| 0.0 | 3.262 | 0.835 | 25.594 | 2.243 | 0.637 | 28.383 |
| −52.0 | 2.968 | 0.824 | 27.772 | 2.104 | 0.633 | 30.065 |
| −78.0 | 2.770 | 0.801 | 28.910 | 1.992 | 0.623 | 31.256 |
| −118.0 | 2.422 | 0.744 | 30.723 | 1.779 | 0.595 | 33.463 |
| −196.0 | 1.682 | 0.578 | 34.379 | 1.286 | 0.500 | 38.915 |
| −255.0 | 1.137 | 0.424 | 37.267 | 0.898 | 0.394 | 43.912 |
| −314.0 | 0.652 | 0.269 | 41.197 | 0.532 | 0.272 | 51.122 |
| −344.0 | 0.263 | 0.196 | 74.537 | 0.222 | 0.209 | 93.956 |
| −431.0 | 0.048 | 0.043 | 89.826 | 0.041 | 0.050 | 121.371 |
| −480.0 | 0.000 | 0.000 | 100.000 | 0.000 | 0.000 | 100.000 |

This reflects the fact that the test displacements began to change rapidly from a pile depth of −344.0 mm, whereas no such inflection point appears in the numerical results. This is believed to be attributable to the following:

- For the ground used in the testing process, the properties of the soil varied by height; in the numerical analysis, by contrast, the ground had constant properties throughout. We believe that this introduced an error in the modeled maximum displacement arising from the lack of change in the ground stiffness with depth.
- We further note that the ground used in the experimental assessment was out of the elastic range when the vibration load was applied. For real field ground, the elastic range will be very small, but, for the convenience of analysis, the ground material

model used in the numerical analysis was assumed to be an elastic body. This is also believed to have contributed to an error in the maximum displacement of the pile model.

### 4.5. Numerical Analysis Result

Numerical analysis was conducted to produce a frequency response function (FRF) plot for three-story SDOF and MDOF structures (Figure 11). For the SDOF structure, the natural frequency occurred at approximately 20.6 Hz, with only a slight increase in response observed in the 3–10 Hz range that reached approximately 1.07 dB at 10 Hz. Although it is not possible to express the overall increase in soil resistance shown in Figure 6b, it can be seen from that figure that the effect arising from the natural frequency of the SDOF structure was insignificant in the 3–10 Hz range. In the case of the MDOF structure, the natural frequency has first, second, and third modes at 1.82, 3.41, and 7.82 Hz, respectively, and there were vertices lower than 1 dB at 2.52 and 6.94 Hz. Figure 12 overlaps these results with those shown in Figure 6c. Near 5 Hz, the FRF of the MDOF structure was close to a peak and the ground reaction force increased relative to that on the W/O. By contrast, near 7 Hz—the natural frequency of the ground—the natural frequency of the MDOF structure was reduced and, therefore, the structure generated a lower soil resistance than the W/O structure. These results confirm that the inertia of the superstructure and the natural frequency of the structure alter p-y curves produced taking the soil–structure interaction into account. Therefore, when designing a pile foundation, the frequency of the site and the natural frequency of the structure to be designed must be considered at the same time. This suggests the necessity of additional studies on the effect of structure shape on the inertia force and mass participation rate.

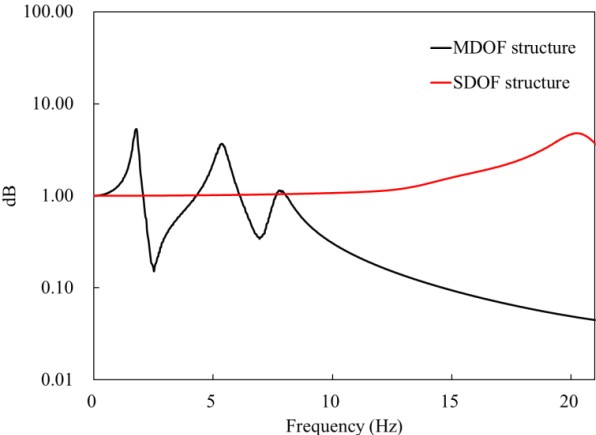

**Figure 11.** FRF for three-story structure.

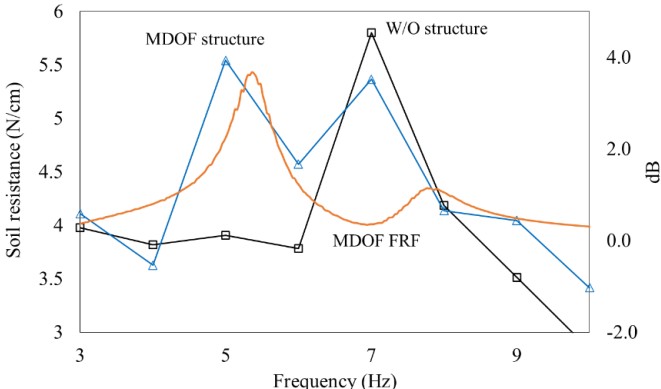

**Figure 12.** Soil resistance w/ MDOF and w/o structures by input frequency overlapped with FRF of a MDOF structure.

## 5. Conclusions

In this study, the p-y curves for superstructures with three modal types in sandy soil were developed using a 1 g shaking table test device. The input frequency and acceleration were shown to be factors influencing the soil–structure interaction and analyzed through numerical modeling. We report the following results:

(1) As the input acceleration increased, the bending moment and pile displacement increased, with the bending moments and pile displacements of the W/O and SDOF models found to be larger than those of the MDOF model. This difference was attributed to the change in the natural period of the upper structure with mode.

(2) The natural frequency of the ground occurred at approximately 7 Hz and coincided with the largest ground reaction force and pile displacement. This phenomenon held for both the SDOF and MDOF models, although there were differences in soil resistance and displacement at other frequencies.

(3) An evaluation of the p-y backbone curves by depth revealed that as the depth increased, the soil resistance increased and the pile displacement decreased, although the pile displacement—and, therefore, the soil resistance—at the ground surface increased sharply in all three cases. This was attributed to the finding that the largest shear force was generated at the head as a result of the loading by the pile cap and upper structure at that position.

(4) A comparison of the results by structure mode revealed that the differences according to mode narrowed as the depth increased. This was attributed to the fact that the ground–pile separation phenomenon occurs when the movement of the ground does not follow the movement of the pile at relatively large displacements in which the input acceleration increases at the top of the pile. As the ground deepens, this segregation phenomenon decreases and, presumably, the load is dispersed through the ground.

(5) Numerical analysis of the model superstructures revealed differences between the natural periods and response spectrum energy distributions of SDOF and MDOF structures with the same weight. The natural frequency of the structure affects the ground and piles, and in order to consider the dynamic load when designing piles, the superstructure, ground, and earthquake waves must all be considered.

(6) In order to remove the limitations of the indoor experiment and apply more accurate SSI, in situ tests and future studies examining the influence of the superstructure are required.

**Author Contributions:** Conceptualization, S.A. and J.J.; methodology, S.A.; software, S.A. and G.P.; validation, S.A. and G.P.; formal analysis, S.A., G.P., H.Y., J.-H.H. and J.J.; investigation, S.A.; resources, S.A. and J.J.; data curation, S.A., G.P. and J.J.; writing—original draft preparation, S.A.; writing—review and editing, J.J.; visualization, S.A.; supervision, J.J.; project administration, J.J.; funding acquisition, J.J. All authors have read and agreed to the published version of the manuscript.

**Funding:** This work is supported by the Korea Agency for Infrastructure Technology Advancement (KAIA) grant funded by the Ministry of Land, Infrastructure and Transport (Grant 21CTAP-C152100-03).

**Institutional Review Board Statement:** Not applicable.

**Informed Consent Statement:** Not applicable.

**Data Availability Statement:** Please contact to corresponding author.

**Acknowledgments:** This work is supported by the Korea Agency for Infrastructure Technology Advancement (KAIA) grant funded by the Ministry of Land, Infrastructure and Transport (Grant 21CTAP-C152100-03).

**Conflicts of Interest:** We declare that we have no financial and personal relationships with other people or organizations that can inappropriately influence our work, there is no professional or other personal interest of any nature or kind in any product, service and/or company that could be construed as influencing the position presented in, or the review of, the manuscript entitled "Evaluation of soil-structure interaction in structure models with shaking table test".

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
