# Peer review of "Evaluation of Soil–Structure Interaction in Structure Models via Shaking Table Test"

_sustainability, doi:10.3390/su13094995_

Round 1
Reviewer 1 Report
The paper investigates an interestimg topic: modeling the soil-structure interaction (SSI) with single-degree-of-freedom (SDOF) and multiple degrees of freedom (MDOF) by comparing shaking table-derived p-y curves with numerical analysis. The paper is well organized and the methodology appropriate. English is also good. Minor revisions are required before approval. 1. Introduction A. This sentence needs to be supported with appropriate reference: "Although more accurate p-y curves based on studies of soil-structure interaction (SSI) under dynamic load conditions have been proposed, such studies have assumed that the superstructure is an SDOF structure" B. The novelty of the paper needs to be explicitly explained. 2. This section has no originality. In needs to be merged with section 3 and reduced. 3. A. "The pile foundation is affected not only by vertical loads from the superstructure but also by lateral dynamic loads produced by the interaction of the superstructure with seismic and wind loads." This sentence needs to be supported by reference. I.e. Mitropoulou CC, Kostopanagiotis C, Kopanos M, Ioakim D, Lagaros ND (2016) Infuence of soil–structure interaction on fragility assessment of building structures. Structures 6:85–98 Forcellini, D. Analytical fragility curves of shallow-founded structures subjected to Soil-Structure Interaction (SSI) effects. Soil Dyn. Earthq. Eng. 2020, 141, 106487. B. This sentence: "The behavior of a pile foundation depends on the properties of the long, middle, and short piles" needs to be explained: is pile lenght the parameter that describes the behaviour of piles? If so, you need to refer to references. I.e. Finn, W.L., 2005. A study of piles during earthquakes: issues of design and analysis. Bulletin of Earthquake Engineering, 3(2), p.141. Forcellini D., 2020 Analytical Fragility Curves of Pile Foundations with Soil-Structure Interaction (SSI), Geosciences 2021, 11, 66. https://doi.org/10.3390/geosciences11020066 Ko Y. and Yang H Deriving seismic fragility curves for sheet-pile wharves using finite element analysis Soil Dynamics and Earthquake Engineering 123 (2019) 265–277. Yang CSW, DesRoches R, Rix GJ. Numerical fragility analysis of vertical-pile-supported wharves in the western United States. J Earthq Eng 2012;16(4):579–94. C. What "actual structures" does mean? Real structures? Please modify. 4. A. Details of the numerical models need to be extensively described. For example, number of elements, material models, boundary conditions, analyses procedures... B. Figure 4 does not show the piles, that are fundamental in this study. Please add a vertical section and a plan view to details their positions. 5. A. You mention "Previous research results have shown that the change in moment decreases at the ground surface", please specify these results. B. Modify "large than" with "larger than" C. This sentence is not clear: "However, for both structures the analytical results have smaller maximum displacements than the experimental results." Probably you mean: However, both structures showed maximum displacements smaller than those resulted from the experimental results. Also, please quantify this difference in %. D. This sentence needs to be clarified and expanded as the main results of the study. "These results confirm that the inertia of the superstructure and the natural frequency of the structure alter p-y curves produced taking the SSI into account" Conclusion I suggest to merge the resulted points to the previous section and write a more synthetic and general conclusion that expresses the originality and the applications.Author Response
Please see the attachment

Reviewer 2 Report
This paper models the soil-structure interaction in seismic design via shaking table tests and numerical models incorporating single-degree-of-freedom and multiple-degree-of-freedom structures.
The following comments are provided for the authors to improve the manuscript:
- Lines 117-121: The authors kept saying the satisfaction of the scale factors for the stiffness and thickness of the model pile was limited. I am wondering how convincing the test results could be?
- Line 179: how did the authors determine the loading frequency?
- In figure 10, why the analysis data have such a pronounced divergence with the experimental results? Is this because the experimental results are not that accurate due to the scale effect or because the numerical model is not applicable? It is suggested a Limitation Section should be included before the Conclusions.
